

# Multivariate bias adjustment of high-dimensional climate simulations: The "Rank Resampling for Distributions and Dependences" (R²D²) Bias Correction

Mathieu Vrac

Laboratoire des Sciences du Climat et de l'Environnement (LSCE-IPSL, CNRS)
Centre d'Etudes de Saclay, Orme des Merisiers, 91190 Gif-sur-Yvette, France

*Correspondence to:* Mathieu Vrac (mathieu.vrac@lsce.ipsl.fr)

**Abstract.** Climate simulations often suffer from statistical biases with respect to observations or reanalyses. It is therefore common to correct (or adjust) those simulations before using them as inputs into impact models. However, most bias correction (BC) methods are univariate and so do not account for the statistical dependences linking the different locations and/or physical variables of interest. In addition, they are often deterministic, while stochasticity is frequently needed to investigate climate

uncertainty and to add constrained randomness to climate simulations that do not possess a realistic variability. This study presents a multivariate method of rank resampling for distributions and dependences (R²D²) bias correction allowing to adjust not only the univariate distributions, but also their inter-variable and inter-site dependence structures. Moreover, the proposed R²D² method provides some stochasticity since it can generate as many multivariate corrected outputs as the number of statistical dimensions (i.e., number of grid-cells × number of climate variables) of the simulations to be corrected. It is based

on an assumption of stability in time of the dependence structure – allowing to deal with a high number of statistical dimensions –, that lets the climate model drive the temporal properties and their changes in time. R²D² is applied on temperature and precipitation reanalyses time series with respect to high-resolution reference data over South-East of France (1506 grid-cells). Bivariate, 1506-dimensional and 3012-dimensional versions of R²D² are tested over a historical period and compared to a univariate BC. How the different BC methods behave in a climate change context is also illustrated with an application to

regional climate simulations over the 2071-2100 period. The results indicate that the 1d-BC basically reproduces the climate model multivariate properties, 2d-R²D² is only satisfying in the inter-variable context, 1506d-R²D² strongly improves inter-site properties and 3012d-R²D² is able to account for both. Applications of the proposed R²D² method to various climate datasets are relevant for many impact studies. The perspectives of improvements are numerous, such as introducing stochasticity in the dependence itself, questioning its stability assumption, and accounting for temporal properties adjustment while including

more physics in the adjustment procedures.



# 1 Introduction

Climate change impact studies aim to investigate and understand the consequences of the potential evolutions of the climate system. Impacts can be hydrological with changes of seasonal flows and water resources driven by precipitation changes (e.g., Schneider et al., 2013); agronomical with crop yields perturbed by heat stress and/or rainfall evolutions (e.g., Müller et al., 2010; Wheeler and von Braun, 2013); ecological with plants and animals diversity (in terms of structures or spatial repartitions) modified by future climate conditions (e.g., Araújo and Rahbek, 2006; Tisseuil et al., 2012); among many others. The common point of those impact studies is that they use Global (GCM) or Regional Climate Model (RCM) simulations of different variables over future time periods according to some scenarios as inputs into impact models to project (e.g., hydrological, ecological) consequences of climate changes. However, most of those climate simulations suffer from statistical biases with respect to observations – or more generally reference data. This means that some of their statistical properties, such as mean, variance, distribution, or even temporal, spatial or inter-variables dependence structures may not be representative of what is observed in the reference dataset. Consequently, before employing climate simulations to feed an impact model, it is often mandatory to "bias correct" (or to "adjust") them in order to correct some of their statistical properties (e.g., Christensen et al., 2008; Muerth et al., 2013).

Over the last decade, most of the developed – and therefore applied – bias correction (BC) methods focused on the adjustment of the mean (e.g., Delta method, Xu, 1999), the variance (e.g., simple scaling adjustment, Berg et al., 2012) or more generally on the adjustment of the distribution (e.g., "quantile-mapping", Haddad and Rosenfeld, 1997). Bias adjustments of the whole distribution through quantile-mapping techniques have been quite popular since it allows adjusting not only on the mean and variance but also any quantile of the variable of interest. Hence, many variants have been proposed (e.g., Déqué, 2007; Michelangeli et al., 2009; Kallache et al., 2011; Tramblay et al., 2013; Vrac et al., 2016) and applied in different studies (e.g., Oettli et al., 2011; Colette et al., 2012; Tisseuil et al., 2012; Vigaud et al., 2013). Nevertheless, usually, those approaches only work in a univariate context, which means that they are designed to correct independently one variable at a time, for one location (e.g., grid cell) at a time. So, if the marginal (i.e., univariate) distributions are generally improved, that is closer to the reference ones – even when the BC is used as a preliminary step to downscaling (e.g., Colette et al., 2012; Vrac and Vaittinada Ayar, 2017) –, the inter-sites and inter-variables dependence structures are usually conserved from the climate model simulations to be corrected. Indeed, 1d-BC methods preserving the ranks of the simulations – as it is the case for quantile-mapping approaches – will not correct the copula functions characterizing the dependencies between sites and/or between variables (e.g., Nelsen, 2006; Schoelzel and Friederichs, 2008; Vrac et al., 2011; Bevacqua et al., 2017). Such a preservation of the model dependence can obviously cause some deficiencies in the subsequent impact studies that will use the 1-dimensional bias corrected simulations, if the model copula function is far from that of the references. It is therefore crucial to adjust not only the marginal distributions of the climate simulations, but also their multivariate dependence structures, which is the goal of the present study. A few multivariate methodologies have then be proposed over the last few years (e.g., Bardossy and Pegram, 2012; Piani and Haerter, 2012; Mao et al., 2015; Vrac and Friederichs, 2015; Cannon, 2017; Dekens et al., 2017; Li et al., 2017). Most of these methods can be categorized into one of the two following approaches: the "marginal/dependence" correction





approach and the "successive conditional" correction approach. The "marginal/dependence" BC methods (e.g., "matrix recor-relation" approach in Bardossy and Pegram, 2012; Vrac and Friederichs, 2015; Cannon, 2017; Li et al., 2017) correct separately the 1d-marginal distributions (e.g., one variable at one given location) and the dependence structure, usually under the form of the underlying copula function linking the different marginal distributions. Once those two components of the joint distribution have been corrected, they are reassembled to obtain adjusted data that respect both the univariate and multivariate dependencies. Although they also aim to adjust climate simulations in a multivariate fashion, the "successive conditional" BC methods (e.g., "sequential recorrelation" approach in Bardossy and Pegram, 2012; Piani and Haerter, 2012; Dekens et al., 2017) are based on a slightly different philosophy. They consist first in correcting one given variable (e.g., one variable at one specific location). Then, a second variable (e.g., another variable or another location) is corrected conditionally on the previously corrected variable. The procedure goes on successively for each dimension (variable/location), correcting variable $n$ conditionally on previously corrected variables $(1, ..., n-1)$. However, this approach suffers from two main limitations. First, since at each step the correction is performed conditionally on previously corrected data, this reduces the number of data available for adjusting each simulation. Consequently, the higher the number of variables to correct, the fewer the number of data to perform the bias correction at each successive step, and therefore the less robust the correction. Second, the ordering of the variables in the successive corrections matters: different orderings generally produce different corrections whose the quality (e.g., in terms of multivariate properties) is not equivalent (Piani and Haerter, 2012; Vrac and Friederichs, 2015). For those reasons, the present study deals with the development of a multivariate BC method within the "marginal/dependence" approach. The proposed methodology relies on the "Empirical Copula - Bias Correction" (EC-BC) method (Vrac and Friederichs, 2015) and is intended to fill some of its weaknesses, mainly its lack of flexibility in terms of temporal properties as well as its deterministic aspect. Concerning the time-related weakness, it has to be noted that it is not possible to correct the multidimensional properties of the simulations without changing the rank sequence of the simulations. In other words, any multivariate BC method will necessarily modify the initial rank chronology of the simulated events. For example, the EC-BC method – belonging to the "marginal/dependence" correction family – allows both the corrected 1d-distributions to evolve consistently with the modeled ones and to reproduce the dependence (copula) structure of the references. But the price for this reproduction is that the temporal sequence of the ranks of the corrected data is exactly that of the reference data over the calibration time period, even for an adjustment performed over a future time period (or more generally over a projection/correction time period different from the calibration one). This also implies that this multivariate BC provides deterministic corrections, while some studies pointed out the needs for stochastic corrections or at least the needs for introducing some stochasticity and variability in the BC process (e.g., Wong et al., 2014; Mao et al., 2015; Volosciuk et al., 2017). Hence, the goals of this paper are:

- to propose a multivariate BC (MBC) method for both multi-site and multi-variable simulations;

- to relax the temporal constraints of EC-BC on the corrected data ranks in order to let the climate model drive more the temporal properties and their evolutions and therefore express its own temporal dynamics;

- to introduce some stochasticity in the MBC outputs, or at least to enable the proposed MBC method to provide multiple corrected scenarios.





The proposed method relies on a multivariate Rank Resampling for Distributions and Dependences bias correction and will then be referred to as $R^2D^2$. This paper is organized as follows: the reference and reanalysis datasets used in this study are presented in the next Section 2. The $R^2D^2$ method is then described in Section 3 after some reminders about the copula theory and the EC-BC approach. The design of experiments performed to evaluate $R^2D^2$ over a historical time period is presented in Section 4 and results are provided in Section 5. Section 6 displays an application of $R^2D^2$ to RCM simulations over a future time period. Conclusions, perspectives and discussions are finally given in Section 7.

## 2   Reference and model data

To apply, investigate and evaluate the proposed $R^2D^2$ correction method, a reference dataset and a model dataset to be corrected is needed, as for any BC method. The reference data employed here are daily temperature and precipitation time series from the "Systeme d'Analyze Fournissant les Renseignements Atmosphériques à la Neige" (SAFRAN) reanalysis data (Quintana-Segui et al., 2008) over the south-east region of France $[2^oE, 7.5^oE] \times [42^oN, 45^oN]$ corresponding to 1506 continental grid-cells with an approximate 8×8 km spatial resolution. SAFRAN has been described, validated and employed in many studies (e.g., Quintana-Segui et al., 2008; Lavaysse et al., 2012; Vrac et al., 2012).

The ERA-Interim (hereafter ERA-I, Dee et al., 2011) daily reanalysis data with a $0.75^o$ by $0.75^o$ spatial resolution are used here as model data to be corrected. Temperature at 2 meters (hereafter T2) and precipitation data (hereafter PR) have been extracted for the same spatial domain as for the reference data. The time period from Jan., 1, 1980 to Dec., 31, 2009 is retained for both reference and ERA-I data. Then, each ERA-I grid-cell is first co-located with the SAFRAN grid-cell the closest to its center in order to be associated with a unique reference SAFRAN grid-cell. Next, each BC method to be tested is applied over two distinct periods of the year: one corresponding loosely to "winter" from October 15th to April 14th; the other to "summer" from April 15th to October 14th. For each "season", corrected ERA-I T2 and PR are obtained for the 1995-2009 "evaluation" time period based on BC models calibrated over 1980-1994.

## 3   Reconstructing multi-sites and multi-variables dependence structures

### 3.1   A brief reminder on copulas

In many of the multivariate BC development papers, the notion of "copula functions" is used. Indeed, those functions characterize the rank dependence structure of most multivariate joint distributions (e.g., Nelsen, 2006; Schoelzel and Friederichs, 2008) through the Sklar's theorem (Sklar, 1959). This theorem expresses that any multivariate cumulative distribution function (CDF) can be described by the univariate marginal CDFs of the multivariate random variable and a copula function. The latter is itself a multivariate CDF depicting the statistical dependence of the transformed random variables $U_j = F_{X_j}(X_j)$, where $X_j$ is the j-th variable of the $d$-dimensional random variable $\mathbf{X} = (X_1, \ldots, X_d)^T$ and $F_{X_j}$ the respective marginal CDF. Mathematically,



Sklar's theorem states that any multivariate CDF $F_{\mathbf{X}}$ (e.g., temperature at several stations) can be written as:

$$F_{\mathbf{X}} = C_{\mathbf{X}}\left(F_{X_1}, \ldots, F_{X_d}\right), \tag{1}$$

where $C_{\mathbf{X}}$ is the copula of $\mathbf{X}$. Therefore, any multivariate BC method will necessarily correct the copula of the simulations, explicitly (e.g., Piani and Haerter, 2012; Vrac and Friederichs, 2015) or implicitly (e.g., Cannon, 2017; Dekens et al., 2017).

### 3.2 A brief reminder of the "Empirical Copula - Bias Correction" (EC-BC) approach

The EC-BC approach (Vrac and Friederichs, 2015) takes advantage of the so-called "Schaake Shuffle", described by Clark et al. (2004) and employed in various studies to reconstruct multivariate dependence structures (e.g., Voisin et al., 2010; Verkade et al., 2013; Cannon, 2017, among others). The principle is the following: First, a 1d-BC is performed on each statistical dimension (i.e., for each variable at each location). Second, the univariate bias corrected data are reordered such that their rank time series is identical to that of the reference sample. This univariate shuffling performed separately on each variable allows to reproduce both the temporal, inter-site and inter-variable dependencies of the reference data (see the synthetic example in table 1 of Section 4.c in Vrac and Friederichs, 2015), since it exactly reproduces the empirical copula function of the references. However, if the inter-site and inter-variable dependence structures can be assumed to be stable over time because they can be considered imposed by physical constraints over the region of interest, this is not the case for the temporal structures (or rank chronology of the climate events). For example, rain persistence can shorten or enlarge, or heat waves can increase and/or be more frequent, and seasonality of some specific (temperature or precipitation) events may change, depending on the geographical domain. It is therefore needed to relax the EC-BC temporal constraint to let the climate simulations express their temporal dynamics and evolutions through time. This is the goal of the proposed methodology.

### 3.3 The "Rank Resampling for Distributions and Dependences" ($R^2D^2$) bias correction approach

The $R^2D^2$ method is developed in the "marginal/dependence" context: The main idea of $R^2D^2$ is to take advantage of the Schaake Shuffle as in EC-BC but to relax the constraint of the reproduction of the temporal structure observed in the reference dataset. To do so, a historical time period is used as the calibration time period for which both climate simulations and reference datasets are at one's disposal. The correction is performed over a projection time period (e.,g., a future time period) where only climate simulations are available. The $R^2D^2$ method proceeds as follows:

1. As in EC-BC or any "marginal/dependence" approach, each dimension (variable/location) is first corrected independently from the others by a univariate BC method. In the present study, the CDF-t method is used (e.g., Vrac et al., 2012).

2. Then, a dimension is selected (i.e., one physical variable at one given location) to serve as a "*reference dimension*" for the shuffling. For this specific dimension, the time sequence of the ranks of the 1d-bias corrected data is kept untouched. Note that this sequence is therefore the same as that of the ranks of the simulations to be corrected, at least with a BC method preserving the ranks as it is the case for CDF-t.



3. Next, for each time step $t$ of the projection time period, R$^2$D$^2$ looks for the time step $t^*$ in the calibration time period for which the rank of the reference dimension is the same as the current rank of the reference dimension, i.e., R$^2$D$^2$ searches $t^*$ such that $rank_{dim}^{1d-BC}(t) = rank_{dim}^{ref.data}(t^*)$, where $rank_{dim}^{A}(t)$ is the rank – in the dataset $A$ – of the value taken by the reference dimension $dim$ at time step $t$.

4. Once this time step $t^*$ is found, the time series of the other dimensions (i.e., the other variables at the same location, and all variables at the other locations) are shuffled such that the inter-site and inter-variable rank structures of the reference dataset are reproduced. This means that the rank association found in the reference dataset for time $t^*$ is reproduced for time $t$.

5. Steps 2. to 4. are then repeated successively until each dimension has served as the reference dimension.

Those different steps are expressed in more mathematical and algorithmic ways in the Appendix A.

An example is now given to illustrate the functioning of R$^2$D$^2$. Let's assume that the multivariate bias correction problem of interest concerns $P = 3$ statistical dimensions. Those can be one physical variable for three grid-cells, or three physical variables for one grid-cell. Each dimension is simulated and observed over $N = 4$ consecutive time steps (e.g., days). Of course, in practice, many more variables, grid-cells and time steps can be treated. Let's say that the first step of independent univariate bias corrections wad performed and that the reference and 1d-bias corrected data are those given in table 1, where each second column indicates the ranks of the values in the time series displayed in the previous column. The results provided by R$^2$D$^2$ are given in table 2. First, a reference dimension is selected, starting with $x$ in this illustration, and the 1d-BC time series of this dimension is preserved at this stage. We can note that the first column of "3d-BC (1/3)" in table 2 is therefore the same as that given by the univariate BC of dimension $x$ in table 1. Then, for each time step (i.e., line in those two tables), the rank of the current 1d-BC value is calculated. The time step with the same rank is then searched into the reference data for this dimension (here, $x$, first and second columns in table 1) and the ranks of the other dimensions $y$ and $z$ for this time step are taken to shuffle the 1d-corrections of those two dimensions. For example, for the first time step in table 2, the value $0.7$ of the reference dimension $x$ has rank 3. Looking into table 1, rank 3 is found at the last time step for $x$, and is associated with ranks 4 and 3 for $y$ and $z$ respectively. Therefore, the $x = 0.7$ value is associated with values 1.8 and 2.6, which have ranks 4 and 3 for $y$ and $z$ in the univariate bias correction (table 1) This procedure is then repeated for each time step before changing the reference dimension and rank sequence. R$^2$D$^2$ then provides as many corrections as the total number dimensions – or at least as many as the number of reference dimensions employed. Indeed, for practical reasons, it may be needed to apply this algorithm on a reduced number of reference dimensions, therefore reducing the number of corrected outputs. However, whatever the number of reference dimensions or correction scenarios selected, the multivariate corrected data should all have equivalent inter-site and inter-variable copula functions.

Moreover, step 4. assumes that these copula (dependence) functions are stable in time (i.e., stationary) and correspond to those from the reference data. This assumption allows applying the proposed R$^2$D$^2$ method in a high-dimensional context, e.g., more than 3000 statistical dimensions as will be illustrated in the following sections.



In the present study, the CDF-t univariate adjustment method (e.g., Vrac et al., 2012) is used to perform step 1. of the above algorithm. However, other 1d-BC methods can of course be employed instead. Note that the regular quantile-mapping approach (e.g., Déqué, 2007) has also been tested within $R^2D^2$ and similar results were obtained (not shown).

## 4 Design of experiments

This section describes the comparisons that will be performed between different BC methods in the following for evaluating the proposed $R^2D^2$ bias correction methodology.

It is first reminded that, for each tested BC method applied to ERA-Interim reanalyses with SAFRAN data as reference, the calibration period is 1980-1994, while the correction/evaluation period is 1995-2009. Moreover, each calibration/evaluation is performed for daily temperature and precipitation time series on 1506 grid-cells in South-East of France over a 6-month "winter" (October 15th to April 14th) and a 6-month "summer" (April 15th to October 14th).

First of all, the 1-dimensional CDF-t bias correction (e.g., Michelangeli et al., 2009; Vrac et al., 2012) is performed. As it is also the 1d-BC method used in step 1.) of the $R^2D^2$ algorithm, this will allow to evaluate the contribution of the others steps in $R^2D^2$. Then, various configurations of the $R^2D^2$ method are applied and evaluated:

- a 2-dimensional $R^2D^2$ version, where each grid-cell is corrected independently but temperature and precipitation are corrected jointly within each grid-cell. As there are 1506 grid-cells for the present dataset, 1506 applications of 2d-$R^2D^2$ are realized;

- a 1506-dimensional $R^2D^2$ version, where all 1506 grid-cell time series are corrected jointly but separately in temperature and precipitation. Therefore, two 1506d-$R^2D^2$ are realized, one for temperature and one for precipitation;

- a 3012-dimensional version, where temperature and precipitation for all the 1506 grid-cells are corrected jointly. Only one 3012d-BC is needed here.

Note that, as $R^2D^2$ can return as many datasets (or "scenarios") of correction as the number of statistical dimensions, the 2d-$R^2D^2$ versions returns 2 corrected datasets for each grid-cell. However, for the 1506d- and 3012d-$R^2D^2$ versions, a sub-sample of 10 reference dimensions has been selected for each version. Therefore, those versions provide 10 corrected datasets. The 10 reference dimensions have been chosen to cover uniformly the geographical domain for each physical variable.

In the following section 5, the results of those four BC methods (1d, 2d, 1506d, 3012d) as well as the initial dataset to be corrected (ERA-I) are compared according to three different aspects evaluated on the 1995-2009 evaluation period. First, the inter-variable dependence properties are investigated in sub-section 5.5.1. Second, the inter-site dependence structures are compared in sub-section 5.5.2. Finally, although this aspect was not part of the correction design, the temporal properties are also evaluated in sub-section 5.5.3. Indeed, as any multivariate BC method will necessarily modify the initial rank chronology of the simulated events, it is interesting to understand – or at least to quantify – these modifications.



## 5   Results

In this Section, all analyses are realized for the winter season but the main conclusions hold for the summer results that are displayed as Supplementary Materials.

### 5.1   Inter-variable correlations

First, the BC results are compared in terms of inter-variable correlations. To do so, the spearman correlation between tempera-
ture and precipitation time series have been computed for each of the 1506 grid-cells and the resulting maps are shown in Fig. 1.
It is clear that the ERA-I inter-variable correlation map (b) is very different from that of SAFRAN (a), with spearman rank
correlations not only different in their intensities but also in their structure. This strongly examplifies the need of adjustment
of this aspect. The 1d-BC CDF-t method (c) modifies the intensities of those correlations but does not change its structure,

basically conserving that of ERA-I. However, the 2d- and 3012d-$R^2D^2$ methods (d and f, respectively) provide major improve-
ments, since they allow to approximate the temperature-precipitation correlations from SAFRAN (a). Correlation maps from
those two $R^2D^2$ versions are strictly the same, indicating that the 3012d-version is a generalization of the 2d-configuration
(at least in this inter-variable evaluation context). This is not the case for the 1506d-$R^2D^2$ configuration that basically destroys
the inter-variable correlations. Indeed, as in this configuration, temperature and precipitation are adjusted separately and inde-

pendently from each other, the obtained correlations are – by construction – close to 0. This version is designed to take care
of the inter-site dependence but completely disregards and even ruins inter-variable correlations. Note that the 3012d-$R^2D^2$
configuration provides the same correlation map as presented in Fig. 1.f, whatever the reference dimension selected. This is
also true for the 2d- and 1506d-versions where a different reference dimension still generates equivalent correlations.

### 5.2   Spatial correlations

The evaluation is now performed in terms of inter-site and spatial correlation. A Principal Component Analysis (PCA) is first
carried out on each physical variable (i.e., temperature and precipitation) separately but for the whole region of interest (i.e.,
1506 grid-cells). However, before applying the PCA, the daily areal mean has been removed from each daily data. Indeed, the
data present a high day-to-day variability within the region of interest. This strongly impacts the PCA that shows a predominant
Empirical Orthogonal Function (EOF) almost uniform over the region if the areal mean is not removed (not shown). Moreover,

25   as precipitation presents a skewed distribution, all zero precipitation values are put to a non-zero but positive small value $(3.3^{-4})$
and the precipitation PCA is performed on the logarithm of the values (following, e.g., Vrac and Friederichs, 2015), where the
areal mean has been removed. Although the log-precipitation values look more Gaussian than the initial ones, a PCA on those
transformed data should still be interpreted with prudence. This is nevertheless a helpful means to describe spatial modes of
variability. Figures 2 and 3 show the maps of the first EOFs obtained from PCAs applied to temperature or log-precipitation

30   respectively from the different datasets. For both variables, ERA-I first EOF (Figs. 2.b and 3.b) maps are quite dissimilar from
the SAFRAN EOF maps (Figs. 2.a and 3.a). The univariate BC (Figs. 2.c and 3.c) shows similar results as those from ERA-I,
although less pronounced for precipitation (3.c). Concerning the results of the 2-dimensional version of $R^2D^2$ (Figs. 2.d and



3.d), for each grid-cell, they are obtained based on selecting as reference dimension the "other" dimension. In other words, for precipitation the reference dimension is temperature, and for temperature, the reference dimension is precipitation. Indeed, otherwise (i.e., if the reference dimension is the variable of interest), by construction, the spatial structures resulting from the 2d-$R^2D^2$ are exactly the same as those from the 1d-BC presented in figures 2.c and 3.c (not shown). In the present configuration

of the 2d-$R^2D^2$ version, the spatial modes of variability (in Figs. 2.d and 3.d) are different from both the ERA-I and 1d-BC results. They visually look more similar to the SAFRAN results and seem to improve the inter-site dependence structure. But this is not the case for summer results (see Supplementary materials) and they do present some major differences with respect to SAFRAN for both precipitation and temperature in the two seasons. However, the first EOF maps from the 1506-dimensional (2.e and 3.e) and the 3012-dimensional versions (2.f and 3.f) are very close to those from the reference SAFRAN dataset,

indicating a satisfying modelling of the main modes of inter-site variability, both for temperature and (log-) precipitation. This is also confirmed by the eigenvalues and explained variance fractions of the leading EOF for temperature and log-precipitation given in figure 4, as well as by the correlograms (i.e., correlations in function of the distance) displayed in figure 5. In those figures, the results of the 1506d- and 3012d-$R^2D^2$ versions are the same: they stick closely to the SAFRAN eigenvalues and explained variances (figure 4) and reproduce well its correlogram (figure 5), even at long distances. The other datasets show

deviations to SAFRAN more or less pronounced and in agreement with previous figures 2 and 3: ERA-I results are relatively far away from SAFRAN; 1d-BC modifies slightly the spatial properties but stays comparable to ERA-I; 2d-$R^2D^2$ degrades the ERA-I spatial properties (at least in the present configuration). The same conclusions hold for summer (see Supplementary materials) where the 1506d- and 3012d-$R^2D^2$ versions follow SAFRAN spatial properties, although some differences appear between the correlograms at long distances ($> 400$km) especially for temperature.

In order to get more quantification of those results, various Spearman and Pearson correlation matrices was computed for the different datasets (SAFRAN, ERA-I and the BC results) in the evaluation period over the 1506 locations:

– on temperature vs. temperature (resulting in a 1506×1506 spatial correlation matrix);

– on precipitation vs. precipitation (1506 × 1506 spatial correlation matrix);

– on temperature vs. precipitation (1506 × 1506 spatial correlation matrix across the two variables);

– on (temperature, precipitation) vs. (temperature, precipitation) (3012 × 3012 spatial and inter-variable correlation matrix).

The SAFRAN correlation matrix is then subtracted from the correlation matrix of each dataset (ERA-I and the BC results), therefore providing matrices that describe differences of correlations (hereafter referred to as $Diff_{corr}$). The absolute values of the elements of this matrix are then summed and the result – noted $S_{corr}$ – gives a numerical indication on the global quality

of the dataset dependence structure with respect to that of SAFRAN. The values of $S_{corr}$ for each dataset and for the different types of correlations are given in Table 3. The results for the "T2 vs. T2" and "PR vs. PR" correlations are quite similar, showing the good behaviour of the 1506d- and 3012d-BC methods, while it is clear that the 2d-$R^2D^2$ version deteriorates the ERA-I and 1d-BC correlations. For the "T2 vs. PR" correlations, the 2d-BC version is relatively equivalent to the ERA-I and





1d-BC but the 1506d-$R^2D^2$slightly degrades those results, while the 3012-dimensional version is much better. Finally, for the "(T2,PR) vs. (T2,PR)" correlations, the 2d-BC version appears as not adapted, the 1506d-BC improves ERA-I and the 1d-BC but 3012d-$R^2D^2$ provides the best results.

Other analyses of the spatial properties derived for the different BC methods were also performed (e.g., quantile-quantile plots of the daily areal means) but are not provided here since their conclusions were the same as in the presented figures: 1d-BC approximately preserves ERA-I properties that are biased with respect to SAFRAN's; 2d-BC changes the ERA-I spatial statistics but does not necessarily improve them; while 1506d- and 3012d-BC via $R^2D^2$ provides satisfying spatial variability and dependence structures, close to those from SAFRAN.

### 5.3 Temporal correlations

The proposed $R^2D^2$ method is not designed to reproduce, correct or preserve the temporal structure of the simulations to be corrected. However, as any multivariate BC will necessarily modify their rank sequence, it is interesting to understand how strong those modifications are, depending on the $R^2D^2$ version. Hence, Figures 6 and 7 display, for each dataset, the 1-day lag auto-correlation maps over the evaluation period for T2 and PR, respectively. For temperature, the ERA-I data (figure 6.b) have high auto-correlation values between 0.8 and 0.9 in agreement with SAFRAN data (figure 6.a), although the spatial structure is different (not highlighted here). Since the univariate CDF-t method preserves the rank sequence, the 1d-BC results (figure 6.c) have similar auto-correlations. However, the other results (2d, 1506d and 3012d) deeply change the ERA-I auto-correlation values, with a strong reduction from the 2d-BC results (figure 6.d). For the 3012d-$R^2D^2$ version, the auto-correlations depend on the statistical dimension serving as reference. Therefore, five illustrations are provided in panels 6.f1-f5 obtained from five reference dimensions, here corresponding to temperature at five locations. Interestingly, those five locations roughly correspond to the center of the red zones visible in panels 6.f1-f5. Indeed, as the reference dimension preserves the rank sequence of the 1d-BC – and therefore of the model data to be corrected – the same auto-correlation values are found at this specific location. The obtained correlation is somehow also reproduced on a neighborhood more or less extended around this location, and rapidly decreases out of this neighborhood. For precipitation (figure 7), the same behavior is present although less pronounced. Moreover, the ERA-I auto-correlation results (6.b) are not in agreement with SAFRAN (6.a) anymore, and the 1d-BC results (6.c) appear quite different from ERA-I. The changes in behavior of the different BC results come from the precipitation occurrences that are modified both in frequency and in the structure of their sequence (e.g., spells). This is not shown here to constrain this article to a reasonable size but maps of wet and dry spell mean lengths as well as maps of probability of dry day given previous one is wet and the other way around are provided as supplementary Materials for both winter and summer. Nevertheless, in order to have a larger view on the temporal correlation of the different datasets, the mean absolute error (hereafter referred to as MAE) with respect to SAFRAN was computed over the evaluation period for each grid-cell and physical variable, based on the first seven auto-correlation values:

$$MAE = \sum_{n=1}^{7} |\rho_n(D) - \rho_n(SAFRAN)| \qquad (2)$$



where $\rho_n(D)$ is the n-day lag auto-correlation value of the dataset $D$. The resulting values are presented via boxplots – summarizing the spatial variability of the MAE – in Figure 8, and via maps as Supplementary Materials. For temperature (left panels of Figure 8), except for the 2d-BC results that show a degradation of the MAE values compared to those from ERA-I or 1d-BC for both seasons, the conclusions are not exactly the same in winter and in summer. In winter, the MAE results

from the 1506d- and 3012d-BC versions are of lower quality (i.e., higher MAE values) than those from ERA-I. This is not the case in summer where those versions present equivalent or even better (i.e., smaller) MAE values. For precipitation, however, winter and summer results are consistent: all tested BC methods generally improve the ERA-I MAEs – although only slightly for 2d-$R^2D^2$ – and the 1506d- and 3012d-$R^2D^2$MAE are relatively close to those from the 1d-BC that presents the best (i.e., smallest) MAE values.

## 6   Bias correction of RCM simulations

### 6.1   GCM/RCMs runs and scenario

For illustration purposes, in order to evaluate and compare the different BC methods when applied to regional climate simulations over a historical period and in a future climate change context, two RCMs driven by the same GCM are used to provide simulations to be corrected. Those RCMs are (i) the "Weather Research and Forecasting" (WRF) regional climate

model (Skamarock et al., 2008) developed by the National Center for Atmospheric Research, and (ii) the "Rossby Centre regional Atmospheric" model (RCA4, Samuelsson et al., 2011). Both RCMs provide daily simulations at a $0.11^o \times 0.11^o$ spatial resolution over the European domain of the Coordinated Regional Climate Downscaling Experiment (CORDEX, Giorgi et al., 2009; Jacob et al., 2014), and were forced by the "Institut Pierre Simon Laplace" (IPSL) global climate model (Marti et al., 2010; Dufresne et al., 2013) with a "historical" 1950-2005 run, as well as for the 2006-2100 time period under a scenario of

Representative Concentration Pathway associated with a radiative forcing of +8.5W/m2 (RCP8.5) in the year 2100 with respect to the preindustrial period (IPCC, 2013). The calibration of the different BC methods is made over the SAFRAN 1980-2009 time period, and for the same winter and summer seasons as previously. The corrections of the WRF and RCA4 simulations are then performed over 1980-2009 and 2071-2100, and only with 1d-CDF-t, 2d-$R^2D^2$ (for T2 and PR together but for each of the 1506 grid-cells separately) and 3012d-$R^2D^2$. The 1506d-BC version (either on T2 or PR) was not performed in this section

since, in the previous one, it provided either equivalent or lower quality results than the 3012d-$R^2D^2$ version. In the following, results are given for the WRF model in winter but the WRF summer and RCA4 winter and summer results are provided as Supplementary Materials.

### 6.2   Historical evaluations and changes from historical to future climate simulations

This subsection contains a short evaluation of the BC methods applied to the RCM simulations over the 1980-2009 historical

period, as well as an illustration of how the tested BC methods behave and differ from each other in a climate change context, both in terms of inter-variable and inter-site dependencies. First, for each dataset, the inter-variable correlation between T2 and



PR in winter is calculated for each grid-cell for both the historical and future time period. The resulting maps are presented in the left panels of Figure 9. Similarly to the BC of the ERA-I reanalyses, although the inter-variable correlations from WRF and its 1d-BC are quite distinct from the reference ones, the 2d- and 3012d-$R^2D^2$ versions (panels 9.e and 9.g, respectively) provide the same maps as that from SAFRAN (9.i), confirming their performance also on RCM simulations. However, the 2d-
version does not do so well from the spatial perspective, as illustrated in Figure 10 showing the temperature and precipitation correlograms. When driven by the "opposite" variable (i.e., T2 for PR correlograms and PR for T2 correlograms), the 2d-BC correlograms are away from both SAFRAN and RCM data, with a strong fall of correlation as soon as the very short distances (a few km) and a flat behaviour after. As for the 3012d-BC of WRF, its correlogram nicely fits the empirical correlations calculated from SAFRAN for both variables. Regarding the RCM future climate simulations and their bias corrections, right
panels of Figure 9 show the changes (i.e., future - present) of the inter-variable correlations. The 1d-CDF-t method smoothes the RCM changes but preserves their structure, while, as expected, the 2d- and 3012d-BC versions do not present strong changes and therefore tend to provide an inter-variable correlation structure close to that of the SAFRAN data. For the changes in the temperature correlograms (Figure 10.a), the RCM simulations do not present much evolutions from the historical period to 2071-2100, and therefore the 1d-BC does the same. Moreover, neither the 2d-BC nor the 3012d-version show major changes
and so the two versions are consistent with the raw simulations in terms of changes. For precipitation (Figure 10.b), the RCM simulations (solid and dashed black lines) do see some changes in the spatial dependence, and therefore, so does the 1d-BC (green lines). Interestingly, the 3012d-BC (red and orange, superimposed) also captures some changes, although slightly less pronounced. This means that 3012d-$R^2D^2$ allows a change (from historical to future) in the inter-site dependence structure that is consistent with the change provided by the RCM.

## 7   Conclusions and Discussion

### 7.1   Conclusions

A new multivariate bias correction approach was proposed, allowing to correct not only the marginal (univariate) distributions of the climate variables of interest, but also the statistical dependences between the variables, as well as the dependences be-tween the different locations over a given geographical domain. This approach relies on the previously developed "Empirical
Copula - Bias Correction" (EC-BC, Vrac and Friederichs, 2015) method, whose all dependence structures – inter-variable, inter-site and overall temporal - were taken from reference data and exactly reproduced by the EC-BC correction. The sug-gested BC approach is also based on a rank resampling to adjust the copula functions and therefore the dependences of the climate simulations, but this $R^2D^2$ method relaxes the EC-BC temporal constraint to let the climate model of interest express its temporal dynamics. Indeed, $R^2D^2$ is based on the assumption that the inter-site and inter-variable copula (dependence)
functions are imposed by physical constraints over the region of interest and are therefore stable in time. So, their dependence structures can be extracted and reconstructed from reference historical data. However, $R^2D^2$ is explicitly designed to partially respect the changes of the climate model (e.g., from historical to future periods) in terms of temporal (rank) properties. Since these evolutions can be distinct for different physical variables and/or grid-cells, $R^2D^2$ generates multiple bias corrected sce-





narios, which can be considered as a stochasticity describing the possible variability of the different rank chronologies. The assumption of stability of the copula function – which can hence be reproduced from the reference data – allows to apply the multivariate bias correction in a high-dimensional context and at a reasonable computational cost. For example, the dataset generated by 3012d-$R^2D^2$ and analyzed in Section 5 (2734 winter days to be corrected for temperature and precipitation over 1506 grid-cells) was obtained on a regular laptop computer with a 2,2 GHz Intel Core i7 processor and a 8 Go 1600 MHz DDR3 memory. On this computer, for each of the 1506 grid-cells, the application of CDF-t (i.e., calibration and correction) takes about 0.05 second for temperature and 0.01 second for precipitation. Then, for one selected reference dimension, in the 3012-dimensional context, each application of the steps 2.-4. of the $R^2D^2$ algorithm presented in Section 3.3 takes about 15 seconds. Consequently, the whole computation time of the 3012d-$R^2D^2$ version with 10 reference dimensions (and therefore 10 multivariate BC scenarios) took: $1506 \times (0.05 + 0.01) = 90.4$ seconds (for the univariate BC, step 1.) + $10 \times 15 = 150$ seconds (for the 10 iterations of the steps 2.-4.), summing to about 240 seconds = 4 minutes.

$R^2D^2$ was first applied to adjust temperature and precipitation time series from ERA-Interim reanalyses (Dee et al., 2011) with respect to the SAFRAN dataset (Quintana-Segui et al., 2008) under a temporal cross-validation framework on 1506 locations. Different configurations of $R^2D^2$ were compared: a bivariate one (2d-$R^2D^2$) applied to adjust jointly temperature and precipitation but separately for each grid-cell; a 1506-dimensional version (1506d-$R^2D^2$) applied jointly on the 1506 grid-cells but separately for temperature and precipitation; and a 3012-dimensional one (3012d-$R^2D^2$) where the two variables were jointly corrected over the 1506 grid-cells. Those different versions were also compared to the univariate CDF-t bias correction method (e.g., Vrac et al., 2012) and to the raw ERA-I data. The results indicate that the 1d-BC by CDF-t generally reproduces the statistical dependence properties of the data to be corrected, from both the inter-variable, inter-site ad temporal perspectives. Moreover, by construction, if 2d-$R^2D^2$ greatly improves the temperature-precipitation relationship, it does not do so well for inter-site dependences. This is the other way around for the 1506d-$R^2D^2$ that shows satisfying inter-site dependence reconstructions but disregards the inter-variable relationship. However, the 3012d-$R^2D^2$ performs well for both the inter-variable and inter-site properties corrections. Regarding the temporal properties, except for the winter temperature with the tested datasets, most BC versions tend to provide auto-correlation getting slightly closer to SAFRAN's. However, it is worth keeping in mind that none of the multivariate BC versions was designed to adjust the temporal properties.

The different BC versions were then also tested and compared on climate simulations from the WRF and RCA4 Regional Climate Models (RCMs) over the 1980-2009 historical period as well as the 2071-2100 future time period. The 2071-2100 bias corrections was not made to evaluate the methods (since no reference data are available for the future) but rather to illustrate how the different multivariate $R^2D^2$ versions behave in a climate change context. The evaluations over the historical period confirmed the results obtained on ERA-I, indicating a robustness of $R^2D^2$ to the dataset to be corrected.

## 7.2 Perspectives and discussion

The perspectives of this work are both methodological and applied. First, as stated earlier, the variability/stochasticity introduced in the actual $R^2D^2$ version refers only to the timing of the events and does not perturb at all the corrected marginal distributions, neither the spatial dependence between sites and/or variables. More stochasticity could also be included into those





properties. For example, the inference of a parametric modelling of copulas (or more generally of the dependence structures) would provide parameters generally associated with some uncertainty (or confidence intervals). Resampling those parameters based on this uncertainty would then allow to generate "perturbed" copulas consistent with each other, and therefore multivariate corrections that are stochastic in their dependences.

Moreover, based on the results presented in this study, the assumption of conservation of the dependence structure sounds reasonable for the inter-site aspects (Fig. 10) but a bit more questionable for the inter-variable aspects, since the tested RCM shows some evolution of the inter-variable correlation in the future (Fig. 9). A generalization of this type of analysis to many more climate models is therefore needed to assess if the dependence preservation hypothesis is robust. This point can be reformulated as a practical question for multivariate BC developments: Should the (inter-variable and/or -site) dependence
structures evolve from calibration to projection periods? Due to the relative youth of the multivariate BC methods, this is still an open question in the literature that should be further investigated and debated.

    Furthermore, the $R^2D^2$ method only partially preserves the temporal properties of the simulations to be corrected, and all multivariate BC methods necessarily modify the temporal structure and rank chronology of the simulations. If this is indirect for most of them (i.e., when accounting only for inter-site or inter-variable structures), some authors tried specifically to tackle
the question of the temporal properties adjustment, such as Johnson and Sharma (2012) with a nesting 1d-BC model working across multiple time scales, Mehrotra and Sharma (2015) including inter-site dependence, or Mehrotra and Sharma (2016) including multiple meteorological variables. However, no general comparison of the pros and cons of the two approaches has been performed and any BC method for both inter-site, inter-variable and temporal properties will necessarily consist in a trade-off between the temporal modifications brought by the multivariate adjustment and the correction of the temporal aspects,
while respecting their changes from one time period to another.

    More generally, there is not yet a complete intercomparison of the multivariate bias adjustment methods. As the needs for such multivariate methods become crucial for many impact studies, intercomparison exercises are now essential to evaluate the various existing methodologies and to make distinctions, not only between "marginals/dependence" and "successive conditional" correction approaches for example, but also between different methods and assumptions within each approach. If such
an intercomparison study has to be performed first from the climate point of view (i.e., in terms of quality of the corrected climate variables and their various properties), it should also be conducted from the perspective of some specific impacts and impact models, trying to understand how the quality of the bias adjusted simulations transfer into the often non-linear impact model outputs. To do so, applying a high-dimensional $R^2D^2$ (and other methods) to various CMIP5 (and upcoming CMIP6) GCM simulations or to various CORDEX RCM runs would generate useful large datasets of multivariate bias corrected cli-
mate simulations. From the purely climatic point of view, those datasets would provide a corrected ensemble to conduct climate change studies, such as related to detection and attribution questions (e.g., Yiou et al., 2017), to the evolution in risks of compound events (e.g., Bevacqua et al., 2017) or more generally related to understanding of climate changes. From the societal and/or environmental point of view, those ensembles of multivariate corrected simulations would allow to investigate how the correction of the dependence structures might modify the impacts of climate change. This question is quite large and concerns



many domains, such as hydrology, agronomy, ecology, etc., and can have major consequences on adaptation and mitigation strategies.

Finally, if the present study focused on the methodological aspects of the multivariate bias correction, it is worth keeping in mind that any application of a BC method should be performed with some physically-based motivations. Indeed, depending on their intrinsic skills to model specific features, some climate simulations cannot sensibly be corrected, especially in climate change context where artifacts of bias correction may appear while not visible in present climate evaluations (e.g., Maraun et al., 2017). So the development of BC methodologies allowing to include some physics in the adjustment procedure is an important perspective of research, in order to have BC approaches not used as "black boxes" while they should be a support to increase the realism of the climate simulations based on physical knowledge.

## Appendix A: Some more mathematical descriptions of the $R^2D^2$ bias correction method

The multivariate BC method is applied to $P$ statistical dimensions. Those dimensions encompass several physical variables at several grid-cells. For example, if there are $V$ physical variables at each of the $S$ grid-cells, then $P = V \times S$. Each dimension is observed or simulated over $N$ time steps, and in the following $V_p^A(t)$ is the value of the dimension $p$ from the dataset $A$ (reference, raw or corrected simulations) at time $t$.

The $R^2D^2$ method consists in the following steps:

1. Apply separate univariate BC to each dimension. We obtain $P$ univariate time series of $N$ values:
   $\{(V_1^{1dBC}(1), ..., V_1^{1dBC}(N)), ..., (V_P^{1dBC}(1), ..., V_P^{1dBC}(N))\}$;

2. Compute the **time series of ranks** for each 1d-bias corrected dimension p among $(V_p^{1dBC}(1), ..., V_p^{1dBC}(N))$. For example, for dimension $p$, we compute $(\text{rank}(V_p^{1dBC}(1)), ..., \text{rank}(V_p^{1dBC}(N)))$ that will be denoted as $(r_p^{1dBC}(1), ..., r_p^{1dBC}(N))$. Therefore, for each time $t$, we have a $P$-dimensional vector of ranks: $\mathbf{R}^{1dBC}(t) = (r_1^{1dBC}(t), ..., r_P^{1dBC}(t))$;

3. Compute the **time series of ranks** for each dimension p among $(V_p^{ref}(1), ..., V_p^{ref}(N))$ in the reference (calibration) dataset.
   For example, for dimension $p$, we compute $(\text{rank}(V_p^{ref}(1)), ..., \text{rank}(V_p^{ref}(N)))$ that will be denoted as $r_p^{ref}(1), ..., r_p^{ref}(N))$. Therefore, for each time $t$, we have a $P$-dimensional vector of ranks: $\mathbf{R}^{ref}(t) = (r_1^{ref}(t), ..., r_P^{ref}(t))$;

4. Choose one dimension $p$ (e.g., $p = 1$) and $(V_p^{1dBC}(1), ..., V_p^{1dBC}(N))$ as the reference dimension and sequence. Then, for each time $t$ from 1 to $N$ in the projection period:

   (a) Find $t^*$ in the calibration period such that $r_p^{1dBC}(t) = r_p^{ref}(t^*)$ and therefore deduce $\mathbf{R}^{ref}(t^*) = (r_1^{ref}(t^*), ..., r_P^{ref}(t^*))$;

   (b) For time $t$ in the projection period, impose that the $P$-dimensional vector of ranks is
   $\mathbf{R}^{PdBC}(t) = (r_1^{ref}(t^*), ..., r_p^{1dBC}(t), ..., r_P^{ref}(t^*))$;

   (c) For all dimensions $d \neq p$, find the time step $t_d$ such that $r_d^{ref}(t^*) = V_d^{1dBC}(t_d)$. Then, define the $P$-dimensional BC vector at time $t$ as $\mathbf{MBC}(t) = (V_1^{1dBC}(t_1), ..., V_p^{1dBC}(t), ..., V_P^{1dBC}(t_P))$.





So, $\mathbf{MBC} = \{\mathbf{MBC}(t=1), ..., \mathbf{MBC}(t=N)\}$ gathers the $N$ $P$-dimensional vectors. In other words, $\mathbf{MBC}$ is a $P$-dimensional time series of length $N$ and contains the multivariate bias corrected data via R$^2$D$^2$ with dimension $p$ as reference dimension;

5. Repeat steps 4.(a-c) for all dimensions until $P$. This generates $\mathbf{MBC}_{all}$, which gathers $P$ objects $\mathbf{MBC}$ (one per dimension as reference for the shuffling):

$$\mathbf{MBC}_{all} = (\mathbf{MBC}(ref.dim. = 1), ..., \mathbf{MBC}(ref.dim. = P)).$$

*Competing interests.* The author declares that no competing interests are present.

*Acknowledgements.* This work has been partially supported by the ANR-project StaRMIP, the VW-project CE:LLO and the LABEX IPSL project. All computations were made in R. An R package containing functions for the R$^2$D$^2$ approach (with CDFt) should soon be made available on the CRAN website[1] or upon request to the author.

---

[1] http://cran.r-project.org/



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





**Table 1.** One example of 3-dimensional reference data and results from the 1d-bias correction of sample size 4 for illustration of the $R^2D^2$ method. $k()$ indicates the rank within the sample.

| References | | | | | | 1d-BC | | | | | |
|---|---|---|---|---|---|---|---|---|---|---|---|
| $x_R^{(i)}$ | $k(x_R^{(i)})$ | $y_R^{(i)}$ | $k(y_R^{(i)})$ | $z_R^{(i)}$ | $k(z_R^{(i)})$ | $x_{1d}^{(i)}$ | $k(x_{1d}^{(i)})$ | $y_{1d}^{(i)}$ | $k(y_{1d}^{(i)})$ | $z_{1d}^{(i)}$ | $k(z_{1d}^{(i)})$ |
| 0.3 | 1 | 1.1 | 1 | 2.1 | 2 | 0.7 | 3 | 1.3 | 2 | 1.9 | 1 |
| 0.5 | 2 | 1.7 | 3 | 1.8 | 1 | 0.5 | 2 | 1.8 | 4 | 2.9 | 4 |
| 0.9 | 4 | 1.2 | 2 | 3.0 | 4 | 0.2 | 1 | 1.1 | 1 | 2.0 | 2 |
| 0.8 | 3 | 1.9 | 4 | 2.7 | 3 | 0.9 | 4 | 1.4 | 3 | 2.6 | 3 |





**Table 2.** Results of the $R^2D^2$ correction method. As the initial data are 3-dimensional, three time series are provided by $R^2D^2$. $k$ indicates the rank within the sample. Within each 3-dimensional time series, the bold values and ranks indicate the dimension and rank sequence taken as reference.

| 3d-BC (1/3) | | | | | | 3d-BC (2/3) | | | | | | 3d-BC (3/3) | | | | | |
|---|---|---|---|---|---|---|---|---|---|---|---|---|---|---|---|---|---|
| $x^{(i)}$ | $k$ | $y^{(i)}$ | $k$ | $z^{(i)}$ | $k$ | $x^{(i)}$ | $k$ | $y^{(i)}$ | $k$ | $z^{(i)}$ | $k$ | $x^{(i)}$ | $k$ | $y^{(i)}$ | $k$ | $z^{(i)}$ | $k$ |
| **0.7** | **3** | 1.8 | 4 | 2.6 | 3 | 0.9 | 4 | **1.3** | **2** | 2.9 | 4 | 0.5 | 2 | 1.4 | 3 | **1.9** | **1** |
| **0.5** | **2** | 1.4 | 3 | 1.9 | 1 | 0.7 | 3 | **1.8** | **4** | 2.6 | 3 | 0.9 | 4 | 1.3 | 2 | **2.9** | **4** |
| **0.2** | **1** | 1.1 | 1 | 2.0 | 2 | 0.2 | 1 | **1.1** | **1** | 2.0 | 2 | 0.2 | 1 | 1.1 | 1 | **2.0** | **2** |
| **0.9** | **4** | 1.3 | 2 | 2.9 | 4 | 0.5 | 2 | **1.4** | **3** | 1.9 | 1 | 0.7 | 3 | 1.8 | 4 | **2.6** | **3** |





**Table 3.** Values of $S_{corr}$, corresponding to the sum of the absolute values of the elements of the difference correlations matrices for each dataset and for the different types of correlations (see text for details) in winter. Values have to be multiplied by $10^4$. Some methods and applications provide different $S_{corr}$ values depending on the reference dimension. For those cases, the mean $S_{corr}$ value is indicated and the standard deviation is indicated between brackets. Values in bold font indicate the smallest values for the line.

|  | ERA-I | 1d-BC | 2d-$R^2D^2$ | 1506d-$R^2D^2$ (T2 or PR) | 3012d-$R^2D^2$ |
|---|---|---|---|---|---|
| Spearman (T) | 20.1 | 20.1 | 96.4 (+/- 107.9) | **5.4** | **5.4** |
| Pearson (T) | 19 | 18.6 | 96.3 (+/- 109.8) | **4.8** | **4.8** |
| Spearman (PR) | 69 | 40.6 | 73.6 (+/- 46.7) | **5.8** | **5.8** |
| Pearson (PR) | 87.9 | 62.1 | 74.7 (+/- 17.8) | **10.4** | **10.4** |
| Spearman (T2 vs. PR) | 25.8 | 24.5 | 23.4 (+/- 11) | 30.4 (+/- 5.8) | **8** |
| Pearson (T2 vs. PR) | 16.3 | 13.3 | 14.8 (+/- 6.7) | 19.4 (+/- 3.5) | **5.7** |
| Spearman (T and PR) | 140.7 | 109.6 | 216.7 (+/- 83.2) | 71.9 (+/- 11.6) | **27** |
| Pearson (T and PR) | 139.5 | 107.3 | 200.6 (+/- 105.5) | 54 (+/- 7.1) | **26.6** |





**Figure 1.** Inter-variable Spearman correlation maps in winter over the evaluation period from: (a) SAFRAN; (b) ERA-I; (c) 1d-BC (CDF-t); (d) 2d-$R^2D^2$; (e) 1506d-$R^2D^2$ on temperature and 1506d-$R^2D^2$ on precipitation; (f) 3012d-$R^2D^2$.



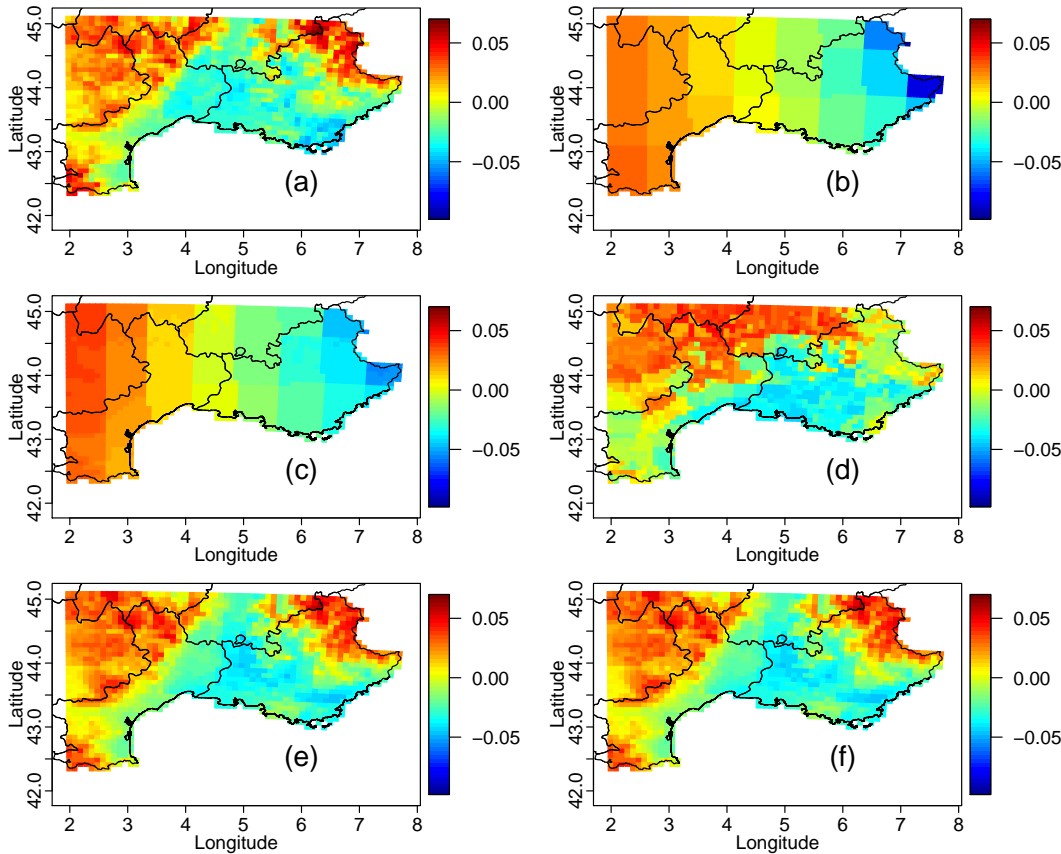

**Figure 2.** Maps of first temperature EOFs in winter over the evaluation period for (a) SAFRAN; (b) ERA-I; (c) 1d-BC (CDF-t); (d) 2d-$R^2D^2$ (with PR as reference dimension); (e) 1506d-$R^2D^2$ (on T2 only); (f) 3012d-$R^2D^2$.





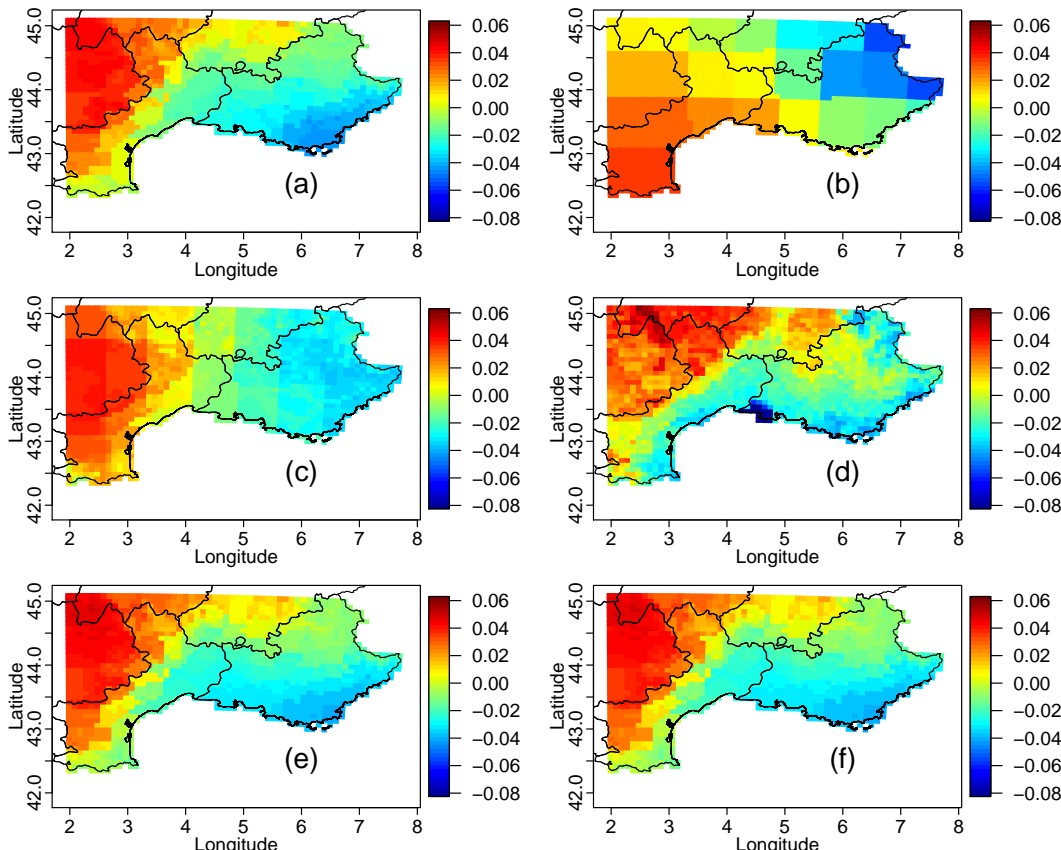

**Figure 3.** Same as Fig. 2 but for precipitation.





**Figure 4.** Eigenvalues (left) and percentage of explained variance (right) of temperature at 2m (first row) and precipitation (second row) in winter over the evaluation period for: SAFRAN (circles); ERAI (dashed); 1d-BC by CDF-t (dotted); 2d-$R^2D^2$ (dashed-dotted); 1506d-$R^2D^2$ (T2 or PR, long dashed); 3012d-$R^2D^2$ (solid line). Note that results of the 1506d- (long dashed) and 3012d-$R^2D^2$ (solid line) versions are the same and are therefore superimposed.



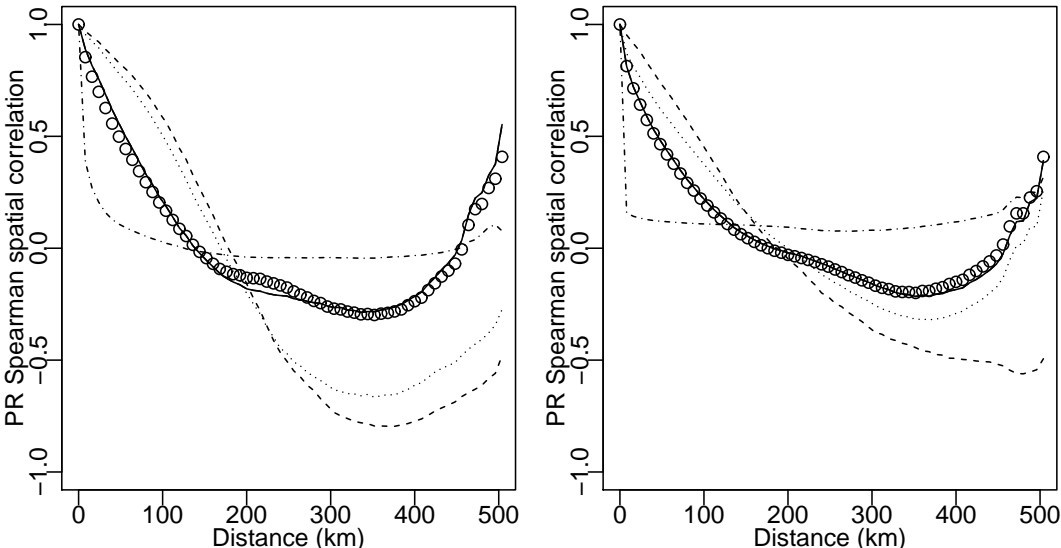

**Figure 5.** Correlograms in winter over the evaluation period for (left) temperature at 2m; (right) precipitation, from SAFRAN (circles); ERAI (dashed); 1d-BC (dotted); 2d-BC (dashed-dotted); 1506d-BC (T2 or PR, long dashed); 3012d-BC (solid line). Note that results of the 1506d-(long dashed) and 3012d-$R^2D^2$ (solid line) versions are the same and are therefore superimposed.





**Figure 6.** Maps of lag-1 day temperature auto-correlations in winter over the evaluation period for (a) SAFRAN; (b) ERA-I; (c) 1d-BC; (d) 2d-$R^2D^2$ (with PR as ref.dim. for each grid-cell); (e) 1506d-$R^2D^2$; (f1-5) 3012d-$R^2D^2$ with 5 different reference temperature locations.





**Figure 7.** Same as Fig. 6 but for precipitation.





**Figure 8.** Boxplots of the MAE values calculated on lag-1 to lag-7 day Pearson correlations in winter for: ERAI; 1d-BC; 2d-BC; 1506d-BC of T2 or PR (example for first reference variable); 3012d-BC with 5 different reference temperature locations. Right: for 2m-temperature; Left: for precipitation; Top row: winter; Bottom row: summer.





**Figure 9.** (left column) Intervariable Pearson correlations between T2 and PR in Winter for each grid cell and (right column) changes in intervariable Pearson correlations from the historical period to the 2071-2100 period; (a-b) for the WRF RCM; (c-d) for its 1d- bias correction with CDFt; (e-f) for its 2d-$R^2D^2$ correction; (g-h) for its 3012d-$R^2D^2$ correction. Panel (i) corresponds to the correlations between T2 and PR for the SAFRAN reference data over the historical period.




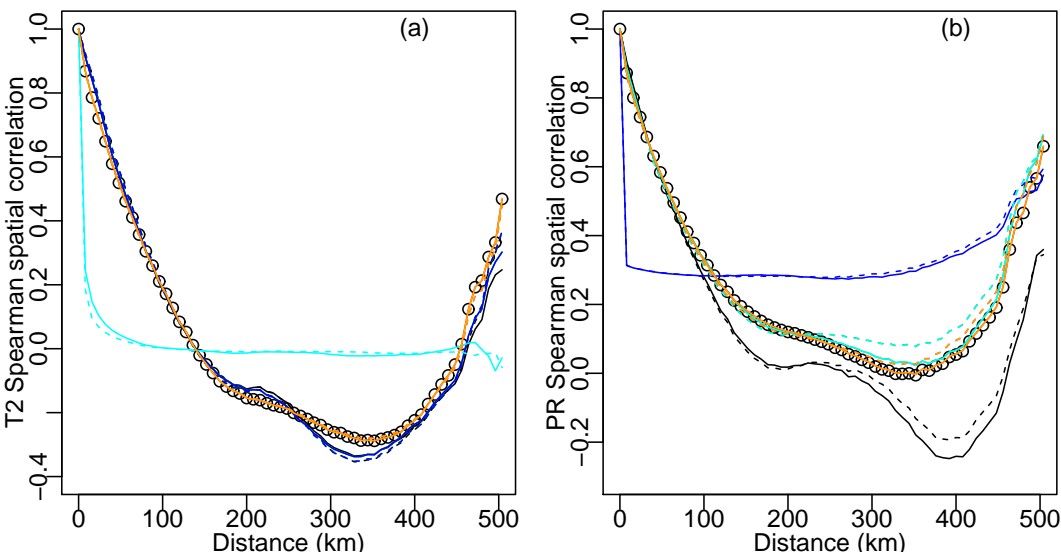

**Figure 10.** Spatial correlograms of temperature (a) and precipitation (b) in winter computed from daily areal mean-removed data. Correlations from SAFRAN are in circles; those from the WRF RCM are in black lines; 1d-BC in green; 2d-$R^2D^2$ driven by temperature in blue; 2d-$R^2D^2$ driven by precipitation in cyan; 3012d-$R^2D^2$ driven by temperature at a given location in red; 3012d-$R^2D^2$ driven by precipitation at the same location in orange. Solide lines indicate results for the historical period, dashed lines for the 2071-2100 period. Note that for temperature results (a), green and blue lines are superimposed. For precipitation (b), green and cyan are superimposed. Red and orange lines are always superimposed for both (a) and (b).