# Peer review of "Multivariate bias adjustment of high-dimensional climate simulations: The "Rank Resampling for Distributions and Dependences" $(R^2D^2)$ Bias Correction"

_Hydrology and Earth System Sciences, 2017_

## Referee Comment (RC1) · Anonymous Referee #1 · 14 Feb 2018

This paper proposes a novel Multivariate bias correction methodology. The author presents this methodology, referred to as R2D2, as the extension of a formerly developed and published methodology (EC-BC). R2D2 is meant to ameliorate some identified weaknesses in EC-BC. These are the "excessive constraint on temporal properties" and the lack of "stochasticity" in the adjustment. These are both laudable endeavours. The results are very interesting and, as I see it, deserving of publication. I have one major comment that, I believe, should be addressed by the author along with a series of minor and/or typographical comments that the author may wish to consider

when editing the manuscript.

Major comment 1: The author tests R2D2 on ERA-interim vs SAFRAN. These are both reanalysis datasets. This may be very interesting per se but is it really a good measure of how R2D2 will perform when applied to RCM data vs reanalysis or observations? I understand that the author does apply the new BC methodology to RCM data but not in a cross validation setting. I would like to understand why he has chosen not to apply R2D2 to RCM data, using SAFRAN as observations, calibrating with one part of the reanalysis data while cross-validating using another. Much like he did with ERA-Interim. In other words, why was ERA-Interim used at all? I find the tests using reanalysis data encouraging to be sure but not as satisfying as a test with RCM data would have been.

Minor comments: Page 1, • Line 10, "allowing to deal" might be "making it possible to deal" or similar?

Page 3, • Line 15, "Whose the quality is not equivalent" could be " with different qualitative results". In any case the sentence might me rewritten. • Line 25, I do not believe this is not the only problem with the Schaake Shuffle. The main problem is not that it is deterministic. The main problem, in my view, is that is unlikely to be robust in time. Why should the rank chronology be the same? This is why I think the authors should cross-validate with a RCM instead of reanalysis data. Page 4, • Line 6, Conclusion, perspective and discussion. The word "Perspective" does not translate identically from French :) . Perhaps "Conclusions, future work and discussion" or simply "Conclusions and discussions". • Line 17, "co-located" could be "regridded" by simple association to nearest neighbour.

Page 6, • Line 2, "R2D2 looks for the time step t∗ in the calibration time period for which the rank of the reference dimension is the same as the current rank of the reference dimension" should add "Please refer to Appendix A for a detailed mathematical description of the R2D2 algorithm". • Line 15, "wad" should be "was" • Line

27, "needed" could be changed to "necessary" • Line 30, If I understand correctly, this last step explains how restrictive the assumption of stationarity is in R2D2. In most BC methods "stationarity" means the "bias" stays the same. Here "stationarity" means the inter variable and inter site structure of the climate stays the same. . . This is a lot more restrictive I think. But these considerations do not matter if a cross-correlation with RCM were performed.

Page 7, • Line 12, Is the parenthesis in "step 1.)" needed here? I found it a bit confusing. Also "others" might be "other" Page 10, • Line 27, "Materials" might be "materials" Page 13, • Line 31, The word "perspectives" here is a bit confusing. See above.

---

## Short Comment (SC1) · 14 Feb 2018

The paper presents an adaptation of the EC-BC (Empirical Copula Bias Correction) method previously introduced by the same author. EC-BC aimed at bias adjusting climate model simulations in a multivariate way (tackling spatial and inter-variable correlations), taking advantage of the Schaake Shuffle. Its main drawback was that the temporal evolution was then forced to follow exactly that of the observations used for the correction. The new R2D2 approach presented here is designed to tackle this problem.

The methodology is clearly described and carefully tested, and sounds attractive. I recommend the publication of the paper. Some points could be clarified.

1) The 1-dimensionnal CDFt bias correction is applied for both temperature and precipitation. Maybe just remind how precipitation occurrence is treated in the case of precipitation, although it is described in the reference (Vrac et al. 2012).

2) Experiment design: the defined winter and summer seasons are 6-month long, which, at least for temperature, still includes some seasonality. Considering months rather than seasons would probably have been better adapted to the distribution estimations, but I am aware that only 15 years for the calibration and validation periods is quite short. Do you think this could lead to better results?

3) Inter-variable correlation: as Spearman correlation is rank based and the correction too, maybe it should be better to check this point for another correlation type.

4) The difficulties linked to the use of PCA in the context of spatial correlation is properly discussed and handled. Could another technique, based on classification for example, have been used instead?

5) The results show that temporal correlations are diversely reproduced, and that the choice of the reference dimension has an impact

a. ERA-I and SAFRAN represent both observations of the same time period, and thus reflect the same temporal evolution. It could be interesting to see how the temporal dynamic is handled by the corrections. The temporal dynamic may be important for impact studies.

b. Can we imagine a way to choose an optimal, or at least some preferred, reference dimension? Could some temporal indicator be of some help? It would probably mean further investigations beyond the scope of the paper, but some thoughts could be added

Typos:

p6 l15: "wad" is written instead of "was"

p7 l12: "of the others steps" instead of "of the other steps"

p13 l19: "inter-site ad temporal perspectives" instead of "and"

——————————————————

---

## Author Comment (AC1) · 22 May 2018

Responses to referees' comments about the manuscript
"Multivariate bias adjustment of high-dimensional climate simulations: The "Rank Resampling for Distributions and Dependences" (R2D2) Bias Correction"
by Mathieu Vrac

First of all, I would like to thank the two referees for their careful reading and constructive comments. I tried to take all of their remarks into account. I think that based on the induced modifications, the manuscript is improved.
A point-by-point response to the reviewers' comments follows, where comments are in black and responses in blue.
* * *
**Anonymous Referee #1**

This paper proposes a novel Multivariate bias correction methodology. The author presents this methodology, referred to as R2D2, as the extension of a formerly developed and published methodology (EC-BC). R2D2 is meant to ameliorate some identified weaknesses in EC-BC. These are the "excessive constraint on temporal properties" and the lack of "stochasticity" in the adjustment. These are both laudable endeavours. The results are very interesting and, as I see it, deserving of publication. I have one major comment that, I believe, should be addressed by the author along with a series of minor and/or typographical comments that the author may wish to consider when editing the manuscript.

Response: I would like to thank the anonymous reviewer. I strongly appreciate the positive feedback. All the referee's comments have been carefully taken into account since they allowed clarifying and improving the manuscript. My responses and modifications brought to the manuscript are indicated below.

Major comment 1:
The author tests R2D2 on ERA-interim vs SAFRAN. These are both reanalysis datasets. This may be very interesting per se but is it really a good measure of how R2D2 will perform when applied to RCM data vs reanalysis or observations? I understand that the author does apply the new BC methodology to RCM data but not in a cross validation setting. I would like to understand why he has chosen not to apply R2D2 to RCM data, using SAFRAN as observations, calibrating with one part of the reanalysis data while cross-validating using another. Much like he did with ERA-Interim. In other words, why was ERA-Interim used at all? I find the tests using reanalysis data encouraging to be sure but not as satisfying as a test with RCM data would have been.

Response:
The ERA-Interim reanalyses were primarily used as data to be corrected because they ensure some consistency with the SAFRAN reference dataset. Indeed, when employing RCM data in a cross-validation context (or more generally when applying a BC method to RCM data over a projection time period different from the calibration time period), the changes in statistical properties (e.g., mean, variance, etc.) from the calibration to the projection time periods can be different for the reference and for the RCM data. Hence, when evaluating the results of a BC method over the projection period, it may be difficult to assess which remaining biases come from the disagreement between reference and RCM changes, and which come from the BC method itself. Using reanalysis data ensures more consistency with the reference and is therefore more appropriate for initial evaluation of a BC method. This is now discussed in the updated manuscript from line 31 of page 4 to line 6 of page 5.

However, the question of the cross-validation for the correction of the RCM simulations is highly relevant. Consequently, the same cross-validation framework as for ERA-I data has also been applied to RCM simulations. For illustration, the inter-variable Spearman correlation maps are presented in the figure below, showing the reference SAFRAN correlations (panel a), those from WRF (b), from the

1D-BC data (c) and finally from the 3012d-$R^2D^2$ (d), both calculated over the "evaluation" period (1995-2009) with a calibration on 1980-1994.

[Figure]

Figure: Inter-variable Spearman correlation maps in winter over the evaluation period (1995-2009) from: (a) SAFRAN; (b) WRF; (c) 1d-BC (CDF-t); (d) 3012d-$R^2D^2$.

It is clear that the conclusions are exactly the same as those provided either without cross-validation with the RCMs, or with cross-validation with ERA-I: WRF inter-variable correlations (b) are quite different from SAFRAN's; 1d-BC basically reproduce the WRF correlation maps; and $R^2D^2$ (both in its 2d and 3012d-versions but only shown for 3012d here) is very close to the SAFRAN correlation map. This is also the case for the EOF analyses in temperature or precipitation performed on the $R^2D^2$ bias corrected dataset with the same cross-validation. Those analyses display the good spatial behaviour of R2D2, i.e., with its first EOFs (in T2 or PR) very similar to those from SAFRAN, as illustrated in the two figures below.

[Figure]

Figure: Maps of first **temperature** EOFs in winter over the evaluation period (1995-2009) from: (a) SAFRAN; (b) WRF; (c) 1d-BC (CDF-t); (d) 3012d-$R^2D^2$.

[Figure]

Figure: Maps of first **precipitation** EOFs in winter over the evaluation period (1995-2009) from: (a) SAFRAN; (b) WRF; (c) 1d-BC (CDF-t); (d) 3012d-$R^2D^2$.

In summary, the cross-validation exercise performed on the bias correction method applied to the RCM simulations does lead to the exact same conclusions as the cross-validation performed on the ERA-Interim reanalyses. Therefore, as this is an important and relevant point, this is now mentioned in the updated manuscript on page 12, lines 25-29.
However, those additional figures are not included in the manuscript and only given here to the attention of the referee and editor.

Minor comments:
Page 1, Line 10, "allowing to deal" might be "making it possible to deal" or similar?
Response: Corrected.

Page 3, Line 15, "Whose the quality is not equivalent" could be " with different qualitative results". In any case the sentence might me rewritten.
Response: Corrected.

Line 25, I do not believe this is not the only problem with the Schaake Shuffle. The main problem is not that it is deterministic. The main problem, in my view, is that is unlikely to be robust in time. Why should the rank chronology be the same? This is why I think the authors should cross-validate with a RCM instead of reanalysis data.

Response:
I think, the reviewer and I agree. Indeed, this is more or less exactly what is written in this paragraph. I was NOT saying that the only or main problem is that it is deterministic (although it can be an issue). Around line 25 of page 3 of the initial submission, I wrote: "*...the price for this reproduction is that the temporal sequence of the ranks of the corrected data is exactly that of the reference data over the calibration time period, even for an adjustment performed over a future time period (or more generally over a projection/correction time period different from the calibration one)*". Indeed, there is no reason why the rank chronology should be the same (actually, there are plenty of reasons for which it should not be the same). This is now clarified in the updated manuscript on page 3, line 27.

Concerning the cross-validation with a RCM, see response to major comment.

Page 4, Line 6, Conclusion, perspective and discussion. The word "Perspective" does not translate identically from French :) . Perhaps "Conclusions, future work and discussion" or simply "Conclusions and discussions".

Response: Corrected, as well as in the title of the subsection 7.2.

Line 17, "co-located" could be "regridded" by simple association to nearest neighbour.
Response: Corrected.

Page 6, Line 2, "R2D2 looks for the time step t* in the calibration time period for which the rank of the reference dimension is the same as the current rank of the reference dimension" should add "Please refer to Appendix A for a detailed mathematical description of the R2D2 algorithm".
Response: To avoid making the description of this step heavier, the suggested mention "*Please refer to Appendix A for a detailed mathematical description of the R2D2 algorithm*" has been added just before starting the description of the different steps, on page 6, lines 7-8 of the updated manuscript.

Line 15, "wad" should be "was"
Response: Corrected.

Line 27, "needed" could be changed to "necessary"
Response: Corrected.

Line 30, If I understand correctly, this last step explains how restrictive the assumption of stationarity is in R2D2. In most BC methods "stationarity" means the "bias" stays the same. Here "stationarity" means the inter variable and inter site structure of the climate stays the same: : : This is a lot more restrictive I think. But these considerations do not matter if a cross-correlation with RCM were performed.
Response: This is true that this part explains that the dependence structure stays the same. However, as indicated in an earlier response, a cross-validation performed with the RCM data provided the exact same conclusions.

Page 7, Line 12, Is the parenthesis in "step 1.)" needed here? I found it a bit confusing. Also "others" might be "other"
Response: The parenthesis has been removed and "others" has been corrected to "other".

Page 10, Line 27, "Materials" might be "materials"
Response: Corrected.

Page 13, Line 31, The word "perspectives" here is a bit confusing. See above.
Response: Corrected.

---

## Author Comment (AC2) · 22 May 2018

Responses to referees' comments about the manuscript
"Multivariate bias adjustment of high-dimensional climate simulations: The "Rank Resampling for Distributions and Dependences" (R2D2) Bias Correction"
by Mathieu Vrac

First of all, I would like to thank the two referees for their careful reading and constructive comments. I tried to take all of their remarks into account. I think that based on the induced modifications, the manuscript is improved.
A point-by-point response to the reviewers' comments follows, where comments are in black and responses in blue.
* * *
**Anonymous Referee #1**

This paper proposes a novel Multivariate bias correction methodology. The author presents this methodology, referred to as R2D2, as the extension of a formerly developed and published methodology (EC-BC). R2D2 is meant to ameliorate some identified weaknesses in EC-BC. These are the "excessive constraint on temporal properties" and the lack of "stochasticity" in the adjustment. These are both laudable endeavours. The results are very interesting and, as I see it, deserving of publication. I have one major comment that, I believe, should be addressed by the author along with a series of minor and/or typographical comments that the author may wish to consider when editing the manuscript.

Response: I would like to thank the anonymous reviewer. I strongly appreciate the positive feedback. All the referee's comments have been carefully taken into account since they allowed clarifying and improving the manuscript. My responses and modifications brought to the manuscript are indicated below.

Major comment 1:
The author tests R2D2 on ERA-interim vs SAFRAN. These are both reanalysis datasets. This may be very interesting per se but is it really a good measure of how R2D2 will perform when applied to RCM data vs reanalysis or observations? I understand that the author does apply the new BC methodology to RCM data but not in a cross validation setting. I would like to understand why he has chosen not to apply R2D2 to RCM data, using SAFRAN as observations, calibrating with one part of the reanalysis data while cross-validating using another. Much like he did with ERA-Interim. In other words, why was ERA-Interim used at all? I find the tests using reanalysis data encouraging to be sure but not as satisfying as a test with RCM data would have been.

Response:
The ERA-Interim reanalyses were primarily used as data to be corrected because they ensure some consistency with the SAFRAN reference dataset. Indeed, when employing RCM data in a cross-validation context (or more generally when applying a BC method to RCM data over a projection time period different from the calibration time period), the changes in statistical properties (e.g., mean, variance, etc.) from the calibration to the projection time periods can be different for the reference and for the RCM data. Hence, when evaluating the results of a BC method over the projection period, it may be difficult to assess which remaining biases come from the disagreement between reference and RCM changes, and which come from the BC method itself. Using reanalysis data ensures more consistency with the reference and is therefore more appropriate for initial evaluation of a BC method. This is now discussed in the updated manuscript from line 31 of page 4 to line 6 of page 5.

However, the question of the cross-validation for the correction of the RCM simulations is highly relevant. Consequently, the same cross-validation framework as for ERA-I data has also been applied to RCM simulations. For illustration, the inter-variable Spearman correlation maps are presented in the figure below, showing the reference SAFRAN correlations (panel a), those from WRF (b), from the

1D-BC data (c) and finally from the 3012d-$R^2D^2$ (d), both calculated over the "evaluation" period (1995-2009) with a calibration on 1980-1994.

[Figure]

Figure: Inter-variable Spearman correlation maps in winter over the evaluation period (1995-2009) from: (a) SAFRAN; (b) WRF; (c) 1d-BC (CDF-t); (d) 3012d-$R^2D^2$.

It is clear that the conclusions are exactly the same as those provided either without cross-validation with the RCMs, or with cross-validation with ERA-I: WRF inter-variable correlations (b) are quite different from SAFRAN's; 1d-BC basically reproduce the WRF correlation maps; and $R^2D^2$ (both in its 2d and 3012d-versions but only shown for 3012d here) is very close to the SAFRAN correlation map. This is also the case for the EOF analyses in temperature or precipitation performed on the $R^2D^2$ bias corrected dataset with the same cross-validation. Those analyses display the good spatial behaviour of R2D2, i.e., with its first EOFs (in T2 or PR) very similar to those from SAFRAN, as illustrated in the two figures below.

[Figure]

Figure: Maps of first **temperature** EOFs in winter over the evaluation period (1995-2009) from: (a) SAFRAN; (b) WRF; (c) 1d-BC (CDF-t); (d) 3012d-$R^2D^2$.

[Figure]

Figure: Maps of first **precipitation** EOFs in winter over the evaluation period (1995-2009) from: (a) SAFRAN; (b) WRF; (c) 1d-BC (CDF-t); (d) 3012d-$R^2D^2$.

In summary, the cross-validation exercise performed on the bias correction method applied to the RCM simulations does lead to the exact same conclusions as the cross-validation performed on the ERA-Interim reanalyses. Therefore, as this is an important and relevant point, this is now mentioned in the updated manuscript on page 12, lines 25-29.
However, those additional figures are not included in the manuscript and only given here to the attention of the referee and editor.

Minor comments:
Page 1, Line 10, "allowing to deal" might be "making it possible to deal" or similar?
Response: Corrected.

Page 3, Line 15, "Whose the quality is not equivalent" could be " with different qualitative results". In any case the sentence might me rewritten.
Response: Corrected.

Line 25, I do not believe this is not the only problem with the Schaake Shuffle. The main problem is not that it is deterministic. The main problem, in my view, is that is unlikely to be robust in time. Why should the rank chronology be the same? This is why I think the authors should cross-validate with a RCM instead of reanalysis data.

Response:
I think, the reviewer and I agree. Indeed, this is more or less exactly what is written in this paragraph. I was NOT saying that the only or main problem is that it is deterministic (although it can be an issue). Around line 25 of page 3 of the initial submission, I wrote: "*...the price for this reproduction is that the temporal sequence of the ranks of the corrected data is exactly that of the reference data over the calibration time period, even for an adjustment performed over a future time period (or more generally over a projection/correction time period different from the calibration one)*". Indeed, there is no reason why the rank chronology should be the same (actually, there are plenty of reasons for which it should not be the same). This is now clarified in the updated manuscript on page 3, line 27.

Concerning the cross-validation with a RCM, see response to major comment.

Page 4, Line 6, Conclusion, perspective and discussion. The word "Perspective" does not translate identically from French :) . Perhaps "Conclusions, future work and discussion" or simply "Conclusions and discussions".

Response: Corrected, as well as in the title of the subsection 7.2.

Line 17, "co-located" could be "regridded" by simple association to nearest neighbour.
Response: Corrected.

Page 6, Line 2, "R2D2 looks for the time step t* in the calibration time period for which the rank of the reference dimension is the same as the current rank of the reference dimension" should add "Please refer to Appendix A for a detailed mathematical description of the R2D2 algorithm".
Response: To avoid making the description of this step heavier, the suggested mention "*Please refer to Appendix A for a detailed mathematical description of the R2D2 algorithm*" has been added just before starting the description of the different steps, on page 6, lines 7-8 of the updated manuscript.

Line 15, "wad" should be "was"
Response: Corrected.

Line 27, "needed" could be changed to "necessary"
Response: Corrected.

Line 30, If I understand correctly, this last step explains how restrictive the assumption of stationarity is in R2D2. In most BC methods "stationarity" means the "bias" stays the same. Here "stationarity" means the inter variable and inter site structure of the climate stays the same: : : This is a lot more restrictive I think. But these considerations do not matter if a cross-correlation with RCM were performed.
Response: This is true that this part explains that the dependence structure stays the same. However, as indicated in an earlier response, a cross-validation performed with the RCM data provided the exact same conclusions.

Page 7, Line 12, Is the parenthesis in "step 1.)" needed here? I found it a bit confusing. Also "others" might be "other"
Response: The parenthesis has been removed and "others" has been corrected to "other".

Page 10, Line 27, "Materials" might be "materials"
Response: Corrected.

Page 13, Line 31, The word "perspectives" here is a bit confusing. See above.
Response: Corrected.
* * *
**Comments from reviewer: S. Parey**

The paper presents an adaptation of the EC-BC (Empirical Copula Bias Correction) method previously introduced by the same author. EC-BC aimed at bias adjusting climate model simulations in a multivariate way (tackling spatial and inter-variable correlations), taking advantage of the Schaake Shuffle. Its main drawback was that the temporal evolution was then forced to follow exactly that of the observations used for the correction. The new R2D2 approach presented here is designed to tackle this problem.
The methodology is clearly described and carefully tested, and sounds attractive. I recommend the publication of the paper. Some points could be clarified.

Response: I would like to thank Dr. Parey for her comments and questions, as well as for her interest in this work. I have tried to clarify as much as possible the different points raised by the reviewer.

1) The 1-dimensionnal CDFt bias correction is applied for both temperature and precipitation. Maybe just remind how precipitation occurrence is treated in the case of precipitation, although it is described in the reference (Vrac et al. 2012).

Response:
For the precipitation variable, CDF-t has been applied based on the relatively common "threshold adaptation" procedure. It consists in defining first a threshold t for which model data below t are put to zero (e.g., Schmidli et al., 2006; Lavaysse et al., 2012). This threshold is chosen such that the frequency of days with model precipitation >t is the same as the frequency of rainy days in the reference (observed) precipitation data set. After this thresholding, only the positive values are corrected by CDF-t with respect to the strictly positive observed values. Other approaches are possible, such as applying a BC model directly on the whole time series including both dry days and rainy days, i.e., without separating the correction methodology into occurrence and intensity (Vrac et al., 2012; Vigaud et al., 2013; Vrac et al., 2016, among others). The latter approach has also been tested for preliminary tests and the results were not sensibly different from those presented in this article (not shown).
This is now clarified on page 7, lines 18-26 of the updated manuscript and the additional references have been added to the "References" Section.

2) Experiment design: the defined winter and summer seasons are 6-month long, which, at least for temperature, still includes some seasonality. Considering months rather than seasons would probably have been better adapted to the distribution estimations, but I am aware that only 15 years for the calibration and validation periods is quite short. Do you think this could lead to better results?

Response:
Indeed, as they still include some seasonality, 6-month long seasons to condition the BC procedures are certainly not the most suited time intervals for practical use or applications.
However, this is a very convenient cutting for testing and illustrating how a newly developed method behaves, which is the main purpose of this article. Indeed, this cutting allows (1) increasing the number of data points (with respect to a monthly cutting) and (2) restricting the number of figures and evaluations that would be multiplied by 2 in case of 3-month seasons, or by 6 with a monthly cutting.
Dr. Parey's question is nevertheless very relevant: I do think indeed that, provided that enough data are available for calibration and projection, the results might be better with a monthly-based approach. However, if seasonality would be potentially improved, this could also introduce some artificial "discontinuities" when going from one month to another, which may be detrimental to some specific applications. Therefore, although this is out of the scope of the present article, it would deserve to be further investigated on specific case studies in future works.
These different points are now discussed at the end of Section 2 "Reference and model data", on page 4, lines 22-30 of the revised article.
3) Inter-variable correlation: as Spearman correlation is rank based and the correction too, maybe it

should be better to check this point for another correlation type.

Response:
The Pearson correlation maps have also been computed. The obtained maps are presented in the figure below.

[Figure]

Figure: Inter-variable Pearson correlation maps in winter over the evaluation period from: (a) SAFRAN; (b) ERA-I; (c) 1d-BC (CDF-t); (d) 2d-R2D2; (e) 1506d-R2D2 on temperature and 1506d-R2D2 on precipitation; (f) 3012d-R2D2.

This figure is similar to Figure 1 of the article but differs only in the fact that it is based on the linear Pearson correlation (as demanded by Dr. Parey) while Fig. 1 is based on Spearman correlation. As can be seen, the conclusions are the same. However, based on Pearson correlations, the larger gap between the ERA-I Pearson correlations (b) and the reference SAFRAN Pearson correlations (a) makes that most maps only use a reduced number of colours, which is not convenient for visual evaluations. This is why the Spearman correlation is chosen in Fig. 1 of the manuscript.
This is now explained at the very beginning of page 9, in sub-section 5.1.
Note that this bigger bias in ERA-I with respect to SAFRAN (when using Pearson rather than Spearman correlations) is due to the fact that the Pearson correlation combines, to some extent, both the biases in temperature and precipitation intensities (i.e., their univariate marginal distributions) and the biases in the rank dependence structure.

4) The difficulties linked to the use of PCA in the context of spatial correlation is properly discussed and handled. Could another technique, based on classification for example, have been used instead?

Response: I appreciate that Dr. Parey points out the efforts made. Of course, other techniques could be used, instead of PCA, to assess the spatial features, as the use of PCA in this context is certainly not the only approach. Dr. parey mentioned "classification" that could indeed be one of them. By clustering (i.e., unsupervised classification) the different raw, BC or reference datasets, spatial characteristics could be derived and compared from one set to another. I do believe that the results would have been relatively equivalent to those presented in sub-section 5.2 (although I admit that I did not test a clustering approach). However, sub-section 5.2 of the article already contains four figures (2,

3, 4 and 5) showing and assessing the behaviour of the R2D2 in a spatial context. Therefore, I considered the number of figures sufficiently high to not include another one.

5) The results show that temporal correlations are diversely reproduced, and that the choice of the reference dimension has an impact
a. ERA-I and SAFRAN represent both observations of the same time period, and thus reflect the same temporal evolution. It could be interesting to see how the temporal dynamic is handled by the corrections. The temporal dynamic may be important for impact studies.

Response:
Sub-section 5.3 "temporal correlations" of the original manuscript is precisely considering this aspect, e.g., stating "*as any multivariate BC will necessarily modify their rank sequence, it is interesting to understand how strong those modifications are, depending on the $R^2D^2$ version*". Therefore, the 1-day lag auto-correlations were investigated for both temperature and precipitation (Figures 6 and 7 respectively). In addition to the 1-day lag auto-correlations, a more integrated evaluation of the temporal properties was also proposed by calculating the mean absolute error (MAE) of each BC dataset with respect to SAFRAN, based on the first seven auto-correlations (see equation (2)). Note also that maps of wet and dry spell mean lengths as well as maps of probability of dry day given previous one is wet and the other way around are provided as supplementary materials for both winter and summer. Therefore, although I agree that other analyses could have been performed instead or in addition, I trust that – at least for this article – the change in temporal dynamics brought by the BC method is sufficiently investigated.

b. Can we imagine a way to choose an optimal, or at least some preferred, reference dimension? Could some temporal indicator be of some help? It would probably mean further investigations beyond the scope of the paper, but some thoughts could be added

Response:
I would like to thank Dr. Parey for this very valuable comment. Indeed, thinking about that, it helped me to define a few more potential extensions and improvements of the $R^2D^2$ method.
    First of all, the selection of an "optimal" reference dimension, or at least some preferential ones, is certainly a necessary future step. The notion of optimality here may however depend on the context of the correction and on the subsequent use of the multivariate bias corrected data. Simple selection methods can however be imagined. For example, a logical choice can be to select the dimension for which the temporal dynamics of the model to be corrected is the most similar to that of the observations over the calibration period. In such a case, that could correspond to the dimension for which the Spearman rank correlation (or an auto-correlation value) is the closest to that of the reference (observational) data. Of course, other selections are possible but this question is left for future work.
    Second, we could also consider a "multivariate" reference vector. For example, instead of relying on a univariate reference dimension, the latter can be a couple (or more generally a n-dimensional vector) of dimensions. This would then ensure that the dependence structure linking those dimensions would be exactly that of the initial model and therefore "preserved" (i.e., not corrected).
    Third, the univariate or multivariate "reference dimension" time series (used to condition the rank resampling in $R^2D^2$) can easily be replaced by physical indices, such as NAO or ENSO indices, coming from the climate model to correct. Hence, by such an approach, $R^2D^2$ would be applied in a conditional process-oriented BC framework.
This is now mentioned in the discussion sub-section on page 16, lines 2-14 of the updated manuscript.

Typos:

p6 l15: "wad" is written instead of "was"
Response: This is now corrected.

p7 l12: "of the others steps" instead of "of the other steps"

Response: Corrected as well.

p13 l19: "inter-site ad temporal perspectives" instead of "and"
Response: This is corrected.